# Efficient and Stable Off-policy Training via Behavior-aware Evolutionary Learning

**Maiyue. Chen**[*]
School of Aritificial Intelligence
Peking University, China
Corrsponding author
mychen@pku.edu.cn

**Guangyi. He**
Department of Medical Bioinformatics
School of Basic Medical Sciences
Peking University, China

**Abstract:** Applying reinforcement learning (RL) algorithms to real-world continuos control problems faces many challenges in terms of sample efficiency, stability and exploration. Off-policy RL algorithms show great sample efficiency but can be unstable to train and require effective exploration techniques for sparse reward environments. A simple yet effective approach to address these challenges is to train a population of policies and ensemble them in certain ways. In this work, a novel population-based evolutionary training framework inspired by evolution strategies (ES) called Behavior-aware Evolutionary Learning (BEL) is proposed. The main idea is to train a population of behaviorally diverse policies in parallel and conduct selection with simple linear recombination. BEL consists of two mechanisms called behavior-regularized perturbation (BRP) and behavior-targeted training (BTT) to accomplish stable and fine control of the population behavior divergence. Experimental studies have shown that BEL not only has superior sample efficiency and stability compared to existing methods but can also produce diverse agents in sparse reward environments. Due to the parallel implementation, BEL also exhibits relatively good computation efficiency, making it a practical and competitive method to train policies for real-world robots.

**Keywords:** Continuous control, Reinforcement learning, Evolution strategies

## 1 Introduction

Recent advances in reinforcement learning (RL) have proven that off-policy deep reinforcement learning (DRL) algorithms possess great potential in solving continuous control problems, especially in terms of sample efficiency [1] [2] [3]. However, off-policy algorithms are also known to be unstable or brittle [4] [5], which to some extent hinders the application of these algorithms to real-world robots. On the other hand, environments with sparse rewards that are common in real-world scenarios also present challenges in terms of exploration.

Along with DRL algorithms, another line of direct policy search methods called evolutionary algorithms (EA) also showed significant success as a result of the improved computational efficiency of modern hardware and clever implementations [6] [7] [8]. Different from DRL algorithms that exploit the sequential structure of the Markov Decision Process (MDP), EA algorithms treat policy search as a black-box optimization problem and utilize a population of randomly perturbed policies to search for better policies. While being less sample efficient, EA methods tend to enjoy properties such as improved stability, efficient parallelization and better diversity due to the utilization of the EA population and EA operations.

Naturally, combining those two paradigms to obtain the best of both worlds has attracted much efforts over the years [9] [10] [11] [12] [13]. The motivation behind these works is to inject the gradient-trained DRL agents into the population and drive the population with policy gradient signals while enjoying the benefits of the EA population. Another perspective is that EA mutation can serve as parameter space noise and improve the exploration ability of RL agents [14]. Such a combination turned out to be very successful, and the resulting hybrid algorithms can beat both of their EA and DRL components.

6th Conference on Robot Learning (CoRL 2022), Auckland, New Zealand.

One of the key factors resulting in the success of the hybrid algorithms is that the EA population provides a way to maintain the behavioral diversity of the policies [15]. Behavioral diversity, which is usually called phenotypic diversity in Evolutionary Robotics (ER) [16] [17], refers to the differences between outputs of the policies within the population here. Intuitively, diversity is helpful for environments with sparse or deceptive reward signals, providing a higher chance of exploring potentially good states [18]. The improved exploration ability also helps the off-policy learning procedure by filling the replay buffer with more diverse transitions [14]. In addition, in a multi-objective scenario, higher behavioral diversity may result in better coverage of different modalities, e.g., training a cheetah robot to run both forward and backward [15].

However, we identified two important problems that remain unsolved by previous EARL methods:

- The first problem is how can the policy be randomly perturbed in a meaningful and controlled manner? A perturbation that is too small would result in no significant changes, while a perturbation that is too large could lead to divergent training.

- The second problem is that although perturbed networks result in different policies, once policies undergo the same RL training process and sample from the sample replay buffer, they may end up being similar and reduce the overall diversity. Ideally, we would like to have a population that is not only diverse after random perturbation but also diverse after RL training.

With the output actions of a policy under different states treated as its behavior, we aim to measure the population's diversity as the mean behavior divergence (action space distance) between each individual policy and the center policy. The main idea of BEL is to maintain the population diversity by solving the above two problems. To solve the first problem, inspired by previous works [19] [12], Behavior-Regularized Perturbation (BRP) is proposed, which can randomly perturb a policy network within a specified behavior divergence range in an online fashion. To solve the second problem, Behavior-Targeted Training (BTT) is proposed to assign a randomly sampled target behavior divergence and inject it into the actor training process, which shares certain similarities with goal-directed exploration [20]. The final proposed training framework is called Behavior-aware Evolutionary Learning (BEL). In BEL, training is conducted in a generational fashion that closely resembles the traditional evolution strategies (ES). In each generation, first, policies are randomly generated by applying BRP to a central mean policy. Then, those offspring policies are trained in parallel with BTT. Finally, all trained policies undergo a weighted linear combination [21] and form the new center policy for the next generation.

## 2  Related works

Model-free off-policy RL algorithms are a class of sample efficient algorithms for continuous control tasks with relatively high dimensional action spaces [1] [22]. Built upon the actor-critic (AC) paradigm, where a pair of actor networks and critic networks are trained simultaneously, the Twin Delayed Deep Deterministic (TD3) algorithm [2] and the Soft Actor-Critic (SAC) algorithm [3] showed great success, and quickly became the go-to algorithms for sample efficient RL training.

Evolutionary algorithms have also gained attention due to the fact that they prove to be competitive alternatives to MDP-based methods [6] [7]. In [6], the authors developed a simplified natural evolution strategies (NES) [23]. The resulting OpenAI ES offers massive scalability while matching the performance of MDP-based methods. In [7], it was shown that the genetic algorithm (GA) was able to evolve networks with four million parameters and achieved competitive performance compared to gradient-based methods.

Combining evolutionary methods and policy gradient-based methods in order to benefit from the best of two worlds soon attracted much attention when [9] first proposed to evolve a population of agents with GA and periodically inject gradient information into the population. Their resulting algorithm ERL outperformed both GA and Deep Deterministic Policy Gradient (DDPG). In [11], the authors managed to use a variant of ES called the cross entropy method (CEM) to evolve the population half of which was composed of EA agents, and the other half was composed of TD3 agents. Their hybrid algorithm CEM-RL turned out to be very competitive and served as a strong baseline for derivative works. Later, [12] pointed out that traditional crossover and mutation operators widely used in GA can be detrimental in the sense that they could destroy learned behaviors. As a remedy, [12] proposed

to conduct crossover with network distillation and mutation with SM-G-SUM [19] which proved to be able to retain learned network behaviors.

## 3 Background

### 3.1 Evolution Strategies (ES)

Evolution strategies (ES) belongs to the gradient-free black-box optimization algorithm family. It was first proposed by Rechenberg [24], and later developed by Schwefel [25]. Mimicking the natural evolution process, ES randomly generates a population of solution vectors (usually with a Gaussian distribution) whose fitness value will be evaluated in a problem-specific manner (for example episodic reward). In its canonical form, ES can be classified into two major versions: the $(\mu, \lambda) - ES$, where $\mu$ parents of the next generation are selected from the current $\lambda$ offspring, and the $(\mu + \lambda) - ES$, where the selection pool contains both the current parents and offspring. The selection operation is usually a simple weighted linear recombination of the population vectors according to their fitness ranks. In this work, we adopt the simplest $(1, \lambda) - ES$ scheme, and model the population with a uniform distribution in terms of behavior divergence.

### 3.2 Twin Delayed Deep Deterministic Policy Gradient (TD3)

As an RL algorithm, TD3 is built upon the Markov Decision Process (MDP) which is described by $< S, A, P, R, \gamma >$. In this formulation, $S$ is the state space, $A$ is the action space, $P$ is the transition function, $R$ is the reward function and $\gamma$ is a discount factor [4]. The goal is to learn an optimal policy function $\pi$ to maximize the expected return $J(\theta) = \mathbb{E}_{s \sim p_\pi, a \sim \pi} [R_0]$. TD3 solves this problem by adopting the actor-critic deterministic policy gradient [26] [1] [2], where a Q-function $Q_\phi$ is learned through the Bellman equation:

$$Q^\pi(s, a) = r + \gamma \mathbb{E}_{s', a'} [Q^\pi(s', a')], \quad a' \sim \pi(s') \tag{1}$$

Then the policy function $\pi_\theta$ is optimized by the deterministic policy gradient:

$$\nabla_\theta J(\theta) = \mathbb{E}_{s \sim p_\pi} \left[ \nabla_a Q^\pi(s, a)|_{a=\pi(s)} \nabla_\theta \pi_\theta(s) \right] \tag{2}$$

For implementation, both $Q_\phi$ and $\pi_\theta$ are optimized with Monte-Carlo estimation with the help of a replay buffer $D$, the loss function of $Q_\phi$ and $\pi_\theta$ are defined as follows:

$$\mathcal{L}_{Q_\phi}^{TD3} = \mathop{\mathbb{E}}_{(s,a,r,s') \sim \mathcal{D}} \left[ \left( Q_\phi(s, a) - \left( r + \gamma \max_{a'} Q_\phi(s', a') \right) \right)^2 \right] \tag{3}$$

$$\mathcal{L}_{\pi_\theta}^{TD3} = - \mathop{\mathbb{E}}_{s \sim \mathcal{D}} \left[ \mathop{\mathbb{E}}_{a \sim \pi_\theta} Q_\phi(s, a) \right] \tag{4}$$

In TD3, three tricks are used to make the above learning process more stable and alleviate the overestimation bias. The first trick is to learn two Q functions and uses the smaller Q-value to form the target Q in eq. (3). The second trick is to delay the target networks updates with regard to Q network updates. The third trick is to add noise to target actions to smooth out Q along changes in action [2].

## 4 Behavior-aware Evolutionary Learning (BEL)

### 4.1 BEL framework

The overall structure of BEL is outlined in fig. 1. In BEL, we maintain a center actor as the population center, and $\lambda$ actors as offspring. In each generation, first, all offspring actors will be initialized around the center actor with Behavior-Regularized Perturbation smoothing (BRP), which will be introduced in detail in section 4.2. Then each offspring actor will undergo Behavior-Targeted Training (BTT) as will be described in section 4.3. After these two processes, all offspring actors will interact with the environment and save their experiences to the replay buffer. Finally, the population selection is conducted with a weighted linear recombination of network parameters according to episodic rewards of the trained offspring actors to form the center actor for next generation. This process is repeated until termination criterion is met.

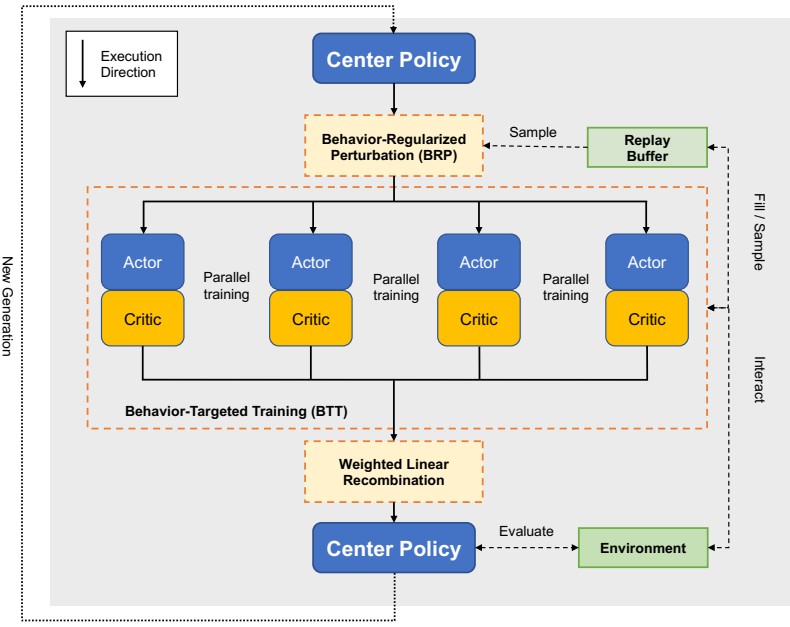

Figure 1: An illustration of the BEL framework.

## 4.2 Behavior-Regularized Perturbation (BRP)

Similar to the SM-G-SUM mutation operator used in [12] and [19], BRP relies on the calculation of the so-called parameter sensitivity with regard to network outputs. Given an actor network $\pi_\theta$ and a batch of transitions $i$, BRP approximately measures how the overall output will vary with regard to small changes of the neural network's weights $\theta$ through the aggregation of backward gradients of each output node $k$ on data batch $i$. For each parameter in $\pi_\theta$, its sensitivity *sens* is calculated by:

$$sens = \sqrt{\sum_k \left( \sum_i |\nabla_\theta \pi_\theta (s_i)_k| \right)^2} \tag{5}$$

A large value of *sens* indicates that the corresponding parameter will lead to a large change of the action output and vice versa. Denoting the overall sensitivity for all parameters as *Sens*, it is then used as the coefficient of the following linear transformation:

$$Vec(\tilde{\pi}_\theta) = Vec(\pi_\theta) + \frac{\delta}{Sens} \tag{6}$$

In eq. (6), $Vec(\pi_\theta)$ means network parameters represented as a one-dimensional vector. $\tilde{\pi}_\theta$ is the perturbed policy network and $\delta$ is a random vector that determines the perturbation magnitude and direction.

Unlike previous methods [27] [12] where $\delta$ is randomly sampled from a constant-scaled Gaussian distribution, the BRP instead tries to adaptively search for a proper $\delta$ within a certain magnitude that can bound the behavior divergence of the perturbed network. This idea is similar to the parameter noise adaptation method in [14].

To measure the behavior divergence in a continuous action space, BRP adopts the simple and widely adopted Euclidean norm as the distance metric [28] [29] [14]:

$$d(\pi_\theta(s), \tilde{\pi}_\theta(s)) = \sqrt{\frac{1}{N} \sum_{k=1}^{N} \mathbb{E}_s \left[ (\pi_\theta(s) - \tilde{\pi}_\theta(s))^2 \right]} \tag{7}$$

Intuitively, this metric is sensitive to large deviations and can tolerate small displacements, which is suitable for common cases. When the action output is not deterministic, such as in SAC, the relative entropy measure is also applicable [30] [16].

---

**Algorithm 1** Behavior-Regularized Perturbation

---

1: **Input:** Population center policy $\pi_\theta$, population size $\lambda$, error bound $\epsilon$, initial magnitude scalar $\delta_{init}$, a batch of states $s$, and $\beta \in [0, 1]$
2: Calculate *Sens* for $\pi_\theta$ according to eq. (5)
3: **for** $i = 1$ **to** $\lambda$ **do**
4:  Sample a target divergence $\Delta_i \sim U_{[0, \Delta_{max}^{BRP}]}$, sample a random direction from $\delta \sim N(0, 1)$
5:  Get initial perturbed network $\tilde{\pi}_i$ with *Sens* according to eq. (6)
6:  **while** $|d(\pi_\theta(s), \tilde{\pi}_i(s)) - \Delta_i| > \epsilon$ **do**
7:   **if** $d(\pi_\theta(s), \tilde{\pi}_i(s)) < \Delta_i$ **then**
8:    $\delta = \frac{1}{\beta}\delta$
9:   **else**
10:    $\delta = \delta\beta$
11:   **end if**
12:   Get perturbed network $\tilde{\pi}_i$ with *Sens* and $\delta$
13:  **end while**
14: **end for**
15: **Output:** Perturbed policies $\{\tilde{\pi}_i | i = 1, ..., \lambda\}$

---

Given a behavior divergence upper bound $\Delta_{max}^{BRP}$, to generate one perturbed network $\tilde{\pi}_i$, BRP first randomly samples a divergence $\Delta_i$, and then conducts a simple iterative line search to find a proper $\delta$, detailed procedure is summarized in algorithm 1. The final output is a set of randomly perturbed policy networks following a uniform distribution in the behavior divergence space. Note that unlike previous implementations which only approximately calculated *Sens*, our implementation precisely calculated *Sens* with the help of Pytorch hooks. Generating five perturbed networks can be done within 0.2 seconds.

### 4.3 Behavior-Targeted Training (BTT)

BRP generates policy networks through random local perturbation, BTT on the other hand generates trained policies that are within a behavior divergence range to the center policy. Consider one offspring policy $\tilde{\pi}_i$ generated by BRP, we would like its behavior divergence after training $\Delta_i^{trained}$ to lie in the range defined by an upper bound: $\Delta_i^{trained} \in \left[0, \Delta_{max}^{BTT}\right]$.

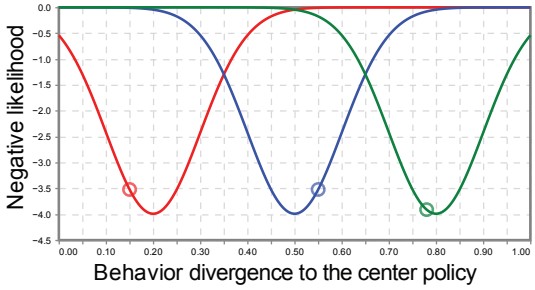

Figure 2: An illustration of the Behavior-targeted training (BTT).

To achieve this goal, we gained inspiration from imitation learning. In each generation of policy gradient training, the actor network aims to optimize two objectives. The first objective is the traditional RL objective as in eq. (4). For the second objective, consider a batch of states $s$ sampled from the replay buffer, the actions of both of the center policy $\pi_\theta$ and the offspring policy $\tilde{\pi}_i$ in those states are calculated as $a_\theta = \pi_\theta(s)$ and $a_i = \tilde{\pi}_i(s)$. Then as depicted in fig. 2, we construct a behavior potential well with a one dimensional Gaussian distribution to force the negative log-likelihood of the Euclidean distance between $a_\theta$ and $a_i$ to stay close to the bottom of the Gaussian whose mean

is defined by a sampled and fixed $\Delta_i^{target}$, standard deviation is defined by a predefined $\sigma_{BTT}$. This process results in the following training objective for BTT:

$$L_{\tilde{\pi}_i}^{BTT} = L_{\tilde{\pi}_i}^{TD3} - \alpha \ln \left[ \frac{1}{\sigma_{BTT}\sqrt{2\pi}} e^{-\frac{\left[ d(\pi_\theta(s), \tilde{\pi}_\theta(s)) - \Delta_i^{target} \right]^2}{2\sigma_{BTT}^2}} \right] \tag{8}$$

Following eq. (8), the policy network will try to simultaneously follow the policy gradient and stay inside the behavior potential well to roughly keep a $\Delta_i^{target}$ divergence to the center policy. $\alpha$ is a hyper-parameter balancing the two objectives. To determine $\Delta_i^{target}$, we simply sample from a uniform distribution as BRP: $\Delta_i^{target} \sim U_{[0, \Delta_{max}^{BTT}]}$. As $\sigma_{BTT}$ directly controls the steepness of the Gaussian distribution, a larger $\sigma_{BTT}$ means less restriction over the policy's divergence and vice versa.

Appendix 0.1 gives more details with regard to weighted linear recombination.

## 5 Experiments

### 5.1 Exploratory studies

As previous studies have shown [11] [14], perturbing Tanh-activated networks is easier than perturbing ReLU-activated networks. As perturbing networks with BRP is straight forward and network architecture agnostic, we conducted a comparative study to see how those two-types of networks respond to BRP. To be specific, we randomly sampled directions and recorded behavior divergence changes along those directions. As in Appendix Figure 1(a), where the x-axis is the percentage of the positive sign in one direction, the y-axis is the magnitude along that direction and the color-scale measures the behavior divergence (the brighter the larger), it is clear that randomly perturbing Tanh-activated networks has a larger chance of inducing significant behavior changes, which explains why Tanh-activated networks are generally favored in perturbation-based methods.

To verify that actors trained by BTT are uniformly distributed as $\Delta_{BTT}^{target}$ is sampled from a uniform distribution, we trained two BEL instances with $\alpha = 0.0$ (without BTT) and $\alpha = 1.0$ (with BTT). From Appendix Figure 1(b), it is obvious that without BTT, the trained policy are quite concentrated. When BTT is applied, the behavior divergences of the population constantly follows the uniform distribution. To further verify that BTT can lead to diverse behaviors, we plotted the state visitation map on the DelayedHalfCheetah-v3 environment to visualize how the offspring policies explore different states. As is shown in Appendix Figure 1(c), the BTT-trained actors (top row) showed relative different state visitation patterns compared with naively-trained actors (bottom row), which suggests that BTT could boost the diversity of the population.

### 5.2 Ablative studies

To verify the effectiveness of BRP, we tested different $\Delta_{max}^{BRP}$ settings on the Walker2d-v3 environment. From Figure 3(a), it is noticeable that BRP not only significantly accelerates the learning process, but also helps avoiding local optimums. As a matter of fact, we find that when BRP is applied, critic networks tend to constantly induce larger training loss throughout training. This phenomenon indicates that BRP indeed brings another level of behavior uncertainty, which forces critics to make better predictions.

An ablation study on the DelayedHalfCheetah-v3 environment is conducted to show that BTT is indeed helpful for exploration .DelayedHalfCheetah-v3 is a modified HalfCheetah-v3 environment where the reward is manually delayed for 20 time steps, making it a difficult sparse reward environment. The proportion of the log-likelihood objective is tuned with $\alpha$. We can observe from Figure 3(b) that when $\alpha$ is set to zero, which means no BTT in the training objective, BEL can't effectively explore. However, on the other hand, when $\alpha$ is too large, actors may also lose performance since their behaviors are over constrained.

As [12] [19] pointed out, many operators in EA are designed for black-box optimization, and can be potentially harmful for neural networks. An experiment comparing weighted linear recombination and distillation-based recombination was designed on the DelayedHalfCheetah-V3 environment, where all parts of BEL are kept the same except for the recombination phase. In this phase, offspring

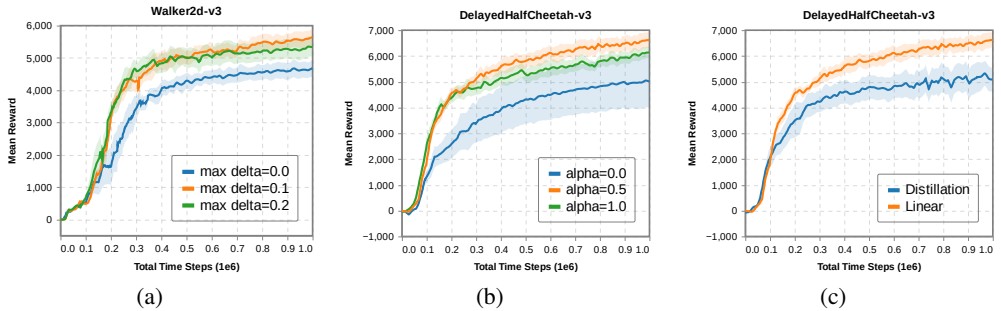

Figure 3: (a) BRP ablation. $\Delta_{max}^{BRP} = 0$ (max delta in plot) corresponds to no BRP (b) BTT ablation. $\alpha = 0$ (alpha in plot) corresponds to no BTT, larger $\alpha$ means more constrained behavior. (c) Recombination ablation.

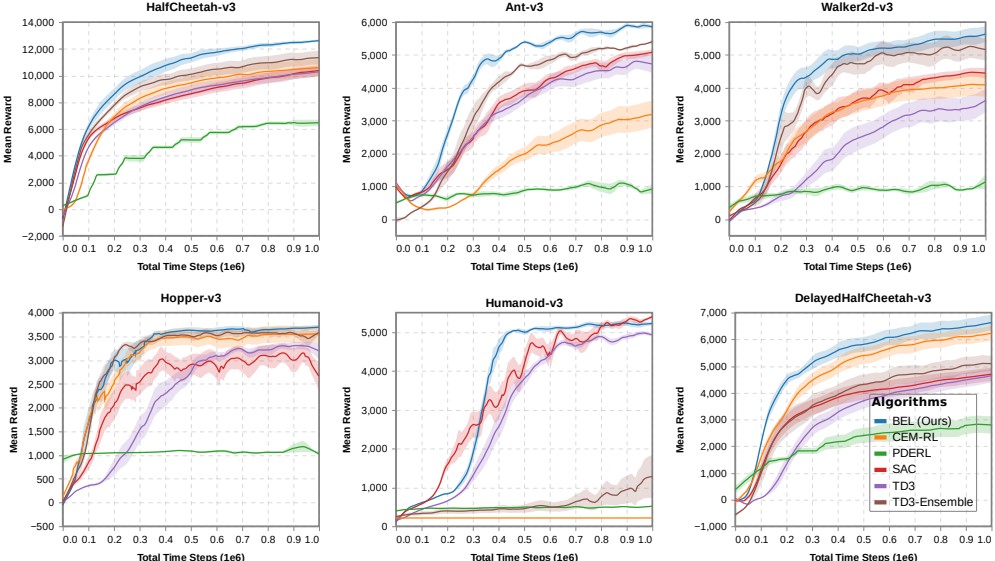

Figure 4: Learning curves on 6 MuJoCo environments in one million time steps.

actors are ranked according to their latest episodic rewards, and then, $\lambda$ offspring actors are treated as teacher networks. Their actions on $N$ sampled observations are recorded as demonstrations. Then, the center actor is treated as the student network to imitate offspring actors. To give different actors different importance, the same set of weights from weighted linear recombination is used:

$$\mathcal{L}(\pi_\theta) = \sum_{k=1}^{N} \sum_{i=1}^{\lambda} \omega_i ||\pi_\theta(s_k) - \tilde{\pi}_i(s_k)||^2 \qquad (9)$$

Much to our surprise, although the distillation-based method seems to learn slightly faster in the early phase, it quickly fell into the local optimum and could hardly make its way out. This experiment showed that though naive linear recombination may break the behavior of the output network to some extent, this kind of behavior uncertainty may result in extra exploration which is beneficial.

## 5.3 Comparison to state-of-the-art RL and EA-RL methods

In this section, the performance of the proposed BEL[1] is compared against pure RL methods including TD3 [2], SAC [3] and TD3-Ensemble as well as other EA-RL methods including [11] and an improved version of ERL which is called PDERL [12]. For TD3, CEM-RL, and PDERL, we used the code

---

[1]Source code for BEL: https://github.com/raymond-myc/BEL

published by the original authors. For SAC, the stable baselines3 library is used. Every algorithm is run on the same machine, and the results we obtained were close to what authors had claimed in the original papers. Five tasks from MuJoCo continuous control benchmark are selected. Swimmer-v3 is excluded since it was found that tuning the reward discount factor to 0.9999 could make all algorithms perform more or less the same, reaching approximately 350 reward. Another DelayedHalfCheetah-v3 environment is constructed by delaying the reward signal for 20 time steps, making it a hard exploration task. Following the convention from other literatures, for all algorithms, their learning curves are aggregated over 10 repeated runs across one million time steps. And the evaluated policies are tested for 10 times. For BEL, the population center policy is used for testing. Note that though BEL trains the population in a parallel fashion, for fair comparison, the total time steps are aggregated for every policy interacting with the environment.

The learning curves are shown in Figure 4, and more detailed results organized in a table with statistical tests are shown in Appendix Table 1.

**Sample efficiency** It can be seen from Figure 4 that BEL turns out to be very competitive against comparing methods in terms of sample efficiency. On the one hand, it can pick up signals faster than other methods, indicating its high sample efficiency. On the other hand, its final best performance outperforms other methods except on Humanoid-v3.

**BEL versus TD3-Ensemble** Since we trained multiple actor-critic pairs in BEL, it is natural to question if the good performance of BEL comes from the ensemble nature. To answer this question, we tested the performance of TD3-ensemble where equal numbers of actor-critic pairs are trained, and all hyper parameters are kept as close as possible. From Figure 4, it is clear that BEL outperforms TD3-ensemble on all tasks.

**Stability** As can be seen from Figure 4 and Appendix Table 1, BEL also generally has smaller standard deviations across runs, even compared to other population based evolutionary methods whose population sizes are larger, this means BEL is very stable. Another phenomenon that suggests BEL's robustness is in the Humanoid-v3 environment, where the naive TD3-Ensemble shared the same learning rate (which is larger than single instance TD3) as BEl, but failed to stably learn.

**Computation efficiency** Since all experiments are conducted on the same machine and all on CPUs, we also recorded the median wall-clock running time of all algorithms. TD3 is the fastest algorithm as it is also the most lightweight algorithm. PDERL ranks the second because not all policies in its population are trained, a great portion of its population are directly evaluated after perturbation. BEL ranks the third among all algorithms, and is generally faster than SAC and CEM-RL. We think BEL reaches a good balance between sample-efficiency and computation overhead.

**Limitations** Though generally good performance can be expected from BEL, it still has the following limitations. First, as multiple networks are trained in parallel, a computation node with a multi-core CPU and relatively large RAM is required. Second, as can be seen from the Humanoid-v3 where BEL does not outperform SAC, it may indicate BRP and BTT do not scale very well as action space dimension grows. Further studies regarding the scalability of BEL need to be conducted.

## 6 Conclusion

In this work, a novel population-based evolutionary training framework for off-policy RL algorithms called BEL is proposed. Exploratory and ablative studies show the effectiveness of BRP and BTT. Benchmark comparisons against other methods show BEL outperforms state-of-the-art RL and EA-RL methods in terms of sample efficiency. The training pipeline is conceptually simple and we offer efficient parallel implementation. Along with the improved stability and exploration ability, we believe BEL can serve as a competitive training method for real-world robot learning with off-policy RL algorithms.

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
