# OpenReview forum: "Efficient and Stable Off-policy Training via Behavior-aware Evolutionary Learning"
_robot-learning.org/CoRL/2022/Conference — CoRL 2022 Poster_

### Official Review · Reviewer_gZYS · 2022-07-26

**Originality:** Very Good
**Technical Quality:** Very Good
**Clarity Of Presentation:** Excellent
**Impact:** 4

**Recommendation:**

Strong Accept: I recommend accepting the paper and will argue for my recommendation even if other reviewers hold a different opinion.

**Summary:**

In the paper "Efficient and Stable Off-policy Training via Behavior-aware Evolutionary Learning", the authors present a novel algorithm called Behavior-aware Evolutionary Learning (BEL). This algorithm enhances an evolutionary strategy algorithm with two different mutation operators that are taking into account the impact of the mutations on the "behaviour" of the policy and encourages diverse behaviours during each generation.
The paper proposes multiple experiments to visualise better the effects of their algorithm as well as three ablation studies. Finally, the authors demonstrate the higher performance of their algorithm against state-of-the-art algorithms in six different environments.

**Issues:**

See above.

**Quality Of The Limitations Section:**

Limitations are addressed clearly

**Reviewer Expertise:**

4: The reviewer is confident but not absolutely certain that the evaluation is correct

**Robotics Focus:**

Relevant but unlikely to deploy to hardware in near future

**Strengths And Weaknesses:**

The paper is overall very well written and easy to understand. The algorithm is easy to understand and is well motivated. The experimental evaluations and ablations demonstrate the benefits of the proposed implementation.

I have three minor issues with the paper:

- What the paper calls "behavioural" diversity is what we would usually call phenotypic diversity in evolutionary robotics (ER), i.e., changes in the policy outputs. Behavioural diversity in ER usually refers to changes in the robot's behaviour, which includes not only changes in the policy outputs but also how these changes interact with the environment. There is for instance a large field of ER called Quality-Diversity (QD) which focuses on promoting behavioural diversity in Policy Search. QD uses "Behavioural descriptors" for this purpose, and I was expecting some sort of connection between the proposed "behaviour-aware" aspect of the algorithm and behavioural descriptors. It was only after reading half of the paper that I realised the terminology mismatch. If this could be clarified in the paper to avoid confusion that would be great.

- The figures could be significantly improved: the titles, legends, and ticks are barely readable. This should be addressed.

- I was unable to find the details on how the weight linear combination is performed. I assume this is relatively trivial (i.e., the weight being proportional to the fitness), but if the paper could specify this it would be great.

**Summary Of Recommendation:**

Apart from the three comments raised above, which could be improved during the rebuttal, the paper is very interesting and would be a nice addition at the conference.

---

> ### Author Response · Authors · 2022-08-22
> **Response to Reviewer gZYS**
>
> - We sincerely thank you for appreciating our work! It is really encouraging for us to hear such positive and constructive feedbacks from the ER community. We'll address the issues in the comments and try to further improve our manuscript as much as possible during the discussion period. The changes made to the manuscript will be colored blue.
> -
>   > It was only after reading half of the paper that I realised the terminology mismatch. If this could be clarified in the paper to avoid confusion that would be great.
>
> 	- (Relevant content has been updated in the revised manuscript)
> 	- Thank you for your in-depth comments, which are very helpful. In BEL, the behavior-aware aspect is indeed essentially the "phenotypic diversity" in ER, as our measure of agent similarity is based on the distance of the network outputs evaluated on the same batch of historical states. Our evaluation method does not take the interaction between agents and environment into consideration, making it less related to the "behavioral descriptors/characteristics" in ER literatures [1] [2].
> -
>   > The figures could be significantly improved: the titles, legends, and ticks are barely readable. This should be addressed.
>
> 	- (Relevant content has been updated in the revised manuscript)
> 	- We apologize for the readability issues of the figures. In the updated manuscript, we have made improvements to them for a better reading experience.
> -
>   > I was unable to find the details on how the weight linear combination is performed. I assume this is relatively trivial (i.e., the weight being proportional to the fitness), but if the paper could specify this it would be great.
>
> 	- (Relevant content has been updated in the revised manuscript)
> 	- We apologize for not being clear about this point. Your assumption is correct, the linear recombination is quite trivial. We first collected the episodic rewards of the actor population as fitness. Then they were weighted with a set of predefined weights as in CMA-ES [3] according to their fitness ranks. Next, a simple weighted linear combination of the network parameters are calculated to form the new central policy.
> - [1] Doncieux, S., & Mouret, J.-B. (2013). Behavioral Diversity with Multiple Behavioral Distances. *IEEE Congress on Evolutionary Computation, 2013 (CEC 2013)*, 1–8. [https://hal.archives-ouvertes.fr/hal-01300703](https://hal.archives-ouvertes.fr/hal-01300703)
> - [2] Gomez, F. J. (2009). Sustaining diversity using behavioral information distance. *Proceedings of the 11th Annual Conference on Genetic and Evolutionary Computation - GECCO ’09*, 113. [https://doi.org/10.1145/1569901.1569918](https://doi.org/10.1145/1569901.1569918)
> - [3] Hansen, N. (2016). The CMA Evolution Strategy: A Tutorial. *ArXiv:1604.00772 [Cs, Stat]*. [http://arxiv.org/abs/1604.00772](http://arxiv.org/abs/1604.00772)

---

### Official Review · Reviewer_4o9R · 2022-07-30

**Originality:** Very Good
**Technical Quality:** Very Good
**Clarity Of Presentation:** Very Good
**Impact:** 3

**Recommendation:**

Weak Accept: I recommend accepting the paper, but will not argue for my recommendation if the majority of other reviewers have a different opinion.

**Summary:**

This paper presents Behavior-aware Evolutionary Learning (BEL), which combines off-policy reinforcement learning with evolutionary strategies (ES) for policy optimization. BEL consists of two parts. The first part is Behavior-Regularized Perturbation (BRP), which improves standard gaussian-based perturbations for ES algorithms by considering the sensitivity of the policy's output to parameter value changes. The second part is Behavior-Targeted Training, which ensures that the fine-tuned policies after RL still have desired diversity. With the combination of BRP and BTT, BEL outperforms baselines in achieved reward. The authors further conducted ablation studies to verify design choices.

**Issues:**

I would like to point out some issues about Fig.2 in the paper. Overall, this figure is pretty difficult to understand due to small font sizes and lack of explanation. A couple of more detailed comments:

Fig.2(a) should probably be a standalone figure and appear earlier in the paper (close to methods section).

Fig.2(b) and (c): the axis labels are too small. In addition, they are referred to as 3(a) and 3(b) in the text (Line 166).

Fig.2(e): I’m having difficulty understanding this figure. Looks like the bottom plots (without BTT) is covering the state space better than the top plots (with BTT)? Why are they in different colors?

Other than these, I do not have specific comments about the writing of the paper. It would be great if the authors could show more robotics-relevant experiments in the future as well.


**Quality Of The Limitations Section:**

Limitations are addressed clearly

**Reviewer Expertise:**

4: The reviewer is confident but not absolutely certain that the evaluation is correct

**Robotics Focus:**

Highly relevant to robotics but no hardware experiments

**Strengths And Weaknesses:**

Strengths:
* Nice incorporation of ES and gradient-based RL

Overall, I think this paper presents an effective method to combine Evolutionary Strategies (ES) with gradient-based Reinforcement Learning. The use of TD3 improves the quality of the random perturbations generated by ES. Moreover, TD3 is an off-policy algorithm, which means that it does not require additional environment interaction to train on. Lastly, the incorporation of BRP and BTT makes the algorithm more effective by ensuring that perturbation happens at desired scale, and the fine-tuned policy remains at desired diversity.

* Well-written and easy-to-understand

The paper is well-organized and easy to follow. Although the paper presents solid ideas with very detailed experiments, it does not feel difficult to read, and the contents flow naturally. I do have a few minor comments about certain details, but the overall writing is great.

* Well-designed experiments

The experiments are well designed and justifies the performance, as well as the detailed design choices of the algorithm.

Weaknesses:
* Less related to robotics
The paper presents a general RL algorithm with evaluation on standard Mujoco benchmarks. While the algorithm should be easily adapted to robotics tasks, the evaluation is mostly in simulation. Overall, I think this paper is clearly relevant to robotics. However, additional robot experiments would be desired for a conference like CoRL.


**Summary Of Recommendation:**

I think it’s a nice paper that should be accepted. The only thing that holds me back from a “strong accept” is its lack of real-robot experiments. However, even at the current shape, I think this paper shows sufficient relevance to robotics and deserves to be accepted.

---

> ### Author Response · Authors · 2022-08-22
> **Response to Reviewer 4o9R**
>
> - We sincerely thank you for your supportive and thoughtful feedbacks! We'll address the issues in the comments and try to further improve our manuscript as much as possible during the discussion period. The changes made to the manuscript will be colored blue.
> -
>   > I would like to point out some issues about Fig.2 in the paper. Overall, this figure is pretty difficult to understand due to small font sizes and lack of explanation.
>
> 	- (Relevant content has been updated in the revised manuscript)
> 	- We apologize for the readability issues of the figures. In the updated manuscript, we have made improvements to them for better reading experience.
> -
>   > Fig.2(e): I’m having difficulty understanding this figure. Looks like the bottom plots (without BTT) is covering the state space better than the top plots (with BTT)? Why are they in different colors?
>
> 	- For this specific figure, the top row and the bottom row are not drawn in the same coordinate system, although their axis limits are set to be the same. To be more concrete, for the top plots (with BTT), all states visited by all actors are first collected into a single large array. And then the a UMAP model was fitted and applied to this large array to reduce the dimensionality into 2. The same operation was conducted independently for the bottom row (without BTT). So the absolute areas covered by these visualizations do not reflect the difference between the state visitation coverage across rows. What we would like to convey in this figure is the different state visitation patterns within each row, with the color scale indicating the state visitation density (more frequent visitation corresponds to darker color).
> -
>   > It would be great if the authors could show more robotics-relevant experiments in the future as well.
>
> 	- Thank you for your suggestion! Though we currently can only test our algorithm via simulation, we do have plans to conduct further experiments on real robot arms.

---

> > ### Comment · Reviewer_4o9R · 2022-08-26
> > **Thanks for the response.**
> >
> > Thanks for the response. The plot is much easier to read now. I would like to keep the original score of "weak accept" as there is no added experiments that increases the paper's relevance to robotics.

---

### Official Review · Reviewer_XDim · 2022-07-30

**Originality:** Good
**Technical Quality:** Fair
**Clarity Of Presentation:** Good
**Impact:** 3

**Recommendation:**

Weak Accept: I recommend accepting the paper, but will not argue for my recommendation if the majority of other reviewers have a different opinion.

**Summary:**

This work presents a direct policy search framework called BEL (Behavior-aware Evolutionary Learning) based on the evolution strategy (ES) approach. The main idea here is to include a policy-gradient step in the generation of the off-spring policies at each generation of the evolution strategy. To ensure that policies do not deviate too much from the parent policy distribution, a regularization step is added. The paper claims that BEL has superior sample efficiency and stability compared to existing methods, and BEL produces diverse agents in sparse reward scenarios. The paper also claims that due to the parallel nature of the ES approach, BEL is a competitive method to train policies for real-world robots.

**Issues:**

In the abstract: "The main idea is to train a population of behaviorally diverse policies..." => Since behavioral diversity is a key element of the paper, spend some time to explain it clearly in the introduction itself, preferably with a example.

Why BEL is sample efficient? What is the intuition behind it? Generally evolutionary strategies are less sample efficient.

- The last paragraph of the introduction is quite vague. There are many unexplained terms introduced here, e.g., behaviour divergence range, behavior, target behavior divergence etc. A concrete example will make this paragrapgh more clear.

- Fig 1 is more complicated than the text. Why not use a sequntial step by step flow diagram than connecting multiple blocks to each other without any clear start or end. Also, a longer caption of a few sentences would also be useful for readers.

- Eq 6: missiong subscript $\theta$ on the left side of the equation.

- Line 139: What is $\mathbf{S}$ ?

- The texts on the plots and figures are extremely small and illegible.

- Fig 2(d): What is RUT in the legend?

**Quality Of The Limitations Section:**

Limitations are not well addressed

**Reviewer Expertise:**

4: The reviewer is confident but not absolutely certain that the evaluation is correct

**Robotics Focus:**

Relevant but unlikely to deploy to hardware in near future

**Strengths And Weaknesses:**

Although the paper picks up an important problem, that is, sample efficient and stable training of policies, it is hard to see how their approach is solving it. From the beginning of the paper itself, many things are either unclear or unexplained and in several places, the authors simply introduce terms/notations, etc. out of nowhere. For instance, from the title of the paper and the abstract, we get the impression that the paper is about maintaining behaviorally diverse policies during the policy search process. But nowhere in the paper do the authors explain what they call a "behavior". Overall the paper lacks a clear intuition behind why their approach works better than the state-of-the-art algorithms like TD3, SAC, etc in standard mujoco tasks. This is a big claim. So it needs clear justification.

**Summary Of Recommendation:**

The paper needs major revision. The paper lacks clear intuition to justify their big claims. Many things are either unclear or unexplained and in several places, the authors simply introduce terms/notations, etc. out of nowhere. Experiments are not convincing enough.

### Post rebbutal update

The authors have addressed my concerns convincingly and improved relevant sections of the paper. Thus I have updated my recommendation to weak accept.

---

> ### Author Response · Authors · 2022-08-22
> **Response to Reviewer XDim (Part 1/2)**
>
> - Thank you for your detailed and constructive feedbacks. We apologize for the lack of explanations and the poor readability of the figures. We'll try our best to clarify things and further improve our manuscript during the discussion phase. We hope our work can serve as an interesting perspective. The changes made to the manuscript will be colored blue.
> -
>   > Overall the paper lacks a clear intuition behind why their approach works better than the state-of-the-art algorithms like TD3, SAC, etc in standard mujoco tasks. This is a big claim. So it needs clear justification.
>
> 	- Since our method trains a population of policies based on the TD3 algorithm, performance improvement is expected due to the ensemble nature of the agent population. BEL does not intent to replace TD3 or SAC as a basic RL algorithm. Instead, BEL offers a way to train a population of diverse agents with TD3 providing policy gradient signals. And our comparative experiment did show that BEL outperforms TD3 and SAC in terms of sample efficiency (significantly higher mean rewards under most tasks).
> -
>   > In the abstract: "The main idea is to train a population of behaviorally diverse policies..." => Since behavioral diversity is a key element of the paper, spend some time to explain it clearly in the introduction itself, preferably with a example.
>
> 	- (Relevant content has been updated in the revised manuscript)
> 	- We apologize for not clearly stating our motivation behind encouraging diversity, we think diversity is helpful in the following points:
> 		- Diversity is helpful for environments with sparse/deceptive reward signals where more diverse population have a higher chance of exploring potentially good states. [2]
> 		- Gathering a diverse set of experiences can help exploring more effectively and quickly, accelerating the learning procedure. [1]
> 		- A diverse set of skills/behaviors is potentially helpful for multi-objective situations, e.g., training a robot to learn to both run forward and backward. [3]
> -
>   > Why BEL is sample efficient? What is the intuition behind it? Generally evolutionary strategies are less sample efficient.
>
> 	- BEL's sample efficiency mainly comes from the improved exploration ability and the improved stability. First, since we aim to maintain the diversity of the population through BRP and BTT, BEL tends to explore different states quicker and fill the buffer with diverse experiences. Second, the ES-based selection mechanism (weighted linear recombination) can naturally reject sub-optimal policies, leading to a stable central policy.
> 	- As a hybrid algorithm of ES and off-policy RL, BEL can well-utilize the sample efficiency brought by the RL part. Traditional ES is indeed less sample efficient due to the fact that its search signal solely comes from local perturbations, which is not the case in BEL.
> -
>   > The last paragraph of the introduction is quite vague. There are many unexplained terms introduced here, e.g., behaviour divergence range, behavior, target behavior divergence etc. A concrete example will make this paragrapgh more clear.
>
> 	- (Relevant content has been updated in the revised manuscript)
> 	- We apologize for the vagueness of this paragraph. The explanation for these terms has been added in the revised manuscript.
> -
>   > Fig 1 is more complicated than the text. Why not use a sequntial step by step flow diagram than connecting multiple blocks to each other without any clear start or end. Also, a longer caption of a few sentences would also be useful for readers.
>
> 	- (Relevant content has been updated in the revised manuscript)
> 	- Thank you for your suggestion, we've reworked this diagram for better readability.
> -
>   > Eq 6: missiong subscript θ on the left side of the equation.
>
> -
>   > Line 139: What is S ?
>
> -
>   > Fig 2(d): What is RUT in the legend?
>
> 	- (Relevant content has been updated in the revised manuscript)
> 	- We're so sorry for these small errors, we've corrected these mistakes in the revised manuscript.
> 	- In line 139, S should be Sens.
> 	- RUT stands for Randomized Uniform Training, which is the former name of Behavior-Targeted Training (BTT).
> -
>   > The texts on the plots and figures are extremely small and illegible.
>
> 	- (Relevant content has been updated in the revised manuscript)
> 	- We apologize for the readability issues of the figures. In the updated manuscript, we have made improvements to them for a better reading experience.

---

> > ### Author Response · Authors · 2022-08-22
> > **Response to Reviewer XDim (Part 2/2)**
> >
> > - [1] Plappert, M., Houthooft, R., Dhariwal, P., Sidor, S., Chen, R. Y., Chen, X., Asfour, T., Abbeel, P., & Andrychowicz, M. (2018). Parameter Space Noise for Exploration. *ArXiv:1706.01905 [Cs, Stat]*. [http://arxiv.org/abs/1706.01905](http://arxiv.org/abs/1706.01905)
> > - [2] Hong, Z.-W., Shann, T.-Y., Su, S.-Y., Chang, Y.-H., & Lee, C.-Y. (2018). Diversity-Driven Exploration Strategy for Deep Reinforcement Learning. *ArXiv:1802.04564 [Cs, Stat]*. [http://arxiv.org/abs/1802.04564](http://arxiv.org/abs/1802.04564)
> > - [3] Parker-Holder, J., Pacchiano, A., Choromanski, K., & Roberts, S. (2020). Effective Diversity in Population Based Reinforcement Learning. *ArXiv:2002.00632 [Cs, Stat]*. [http://arxiv.org/abs/2002.00632](http://arxiv.org/abs/2002.00632)

---

### Official Review · Reviewer_1HqG · 2022-08-01

**Originality:** Very Good
**Technical Quality:** Very Good
**Clarity Of Presentation:** Very Good
**Impact:** 3

**Recommendation:**

Weak Accept: I recommend accepting the paper, but will not argue for my recommendation if the majority of other reviewers have a different opinion.

**Summary:**

The paper offers a novel population-based evolutionary training framework for off-policy RL algorithms based on the idea of combining both Evolutionary Algorithm (EA) and Deep Reinforcement Learning (DRL) into a single training framework due to the recent promising results when combining and exploiting the advantages of both EQ and DRL methods.

The paper first addresses two current limitations of hybrid approaches:
 - How to randomly perturb the policy in a meaningful and controlled manner?
 - Perturbed networks may end up being similar and reduce the overall diversity.

The paper suggests solving these two problems by proposing to combine:
Behavior-Regularized Perturbation (BRP) to randomly perturb a policy network within a specified behavior divergence range online and
Behavior-Targeted Training (BTT) to randomly assign a sampled target behavior divergence and inject it into the actor training process.

The proposed framework shows the benefits over baselines.

**Issues:**

- computational expensive and the author proposes parallel implementation which shows how solve the problem.

**Quality Of The Limitations Section:**

Limitations are addressed clearly

**Reviewer Expertise:**

1: The reviewer's evaluation is an educated guess

**Robotics Focus:**

Highly relevant to robotics but no hardware experiments

**Strengths And Weaknesses:**

Strengths:
- well-written, clear, and easy to follow
- results are good compared to baselines

Weaknesses
- computational expensive

**Summary Of Recommendation:**

Can you provide more reasons and explanations on why BRP adopts the Euclidean norm and not other distance measures?

Post rebuttal update

The authors answered my question with related references, which convinced me, so I kept my score weak accept.

---

> ### Author Response · Authors · 2022-08-22
> **Response to Reviewer 1HqG**
>
> - We sincerely thank you for your appreciation and constructive feedbacks! We've further revised our manuscript to offer more explanations and added statistical tests, please do check it out. The changes made to the manuscript will be colored blue.
> -
>   > Can you provide more reasons and explanations on why BRP adopts the Euclidean norm and not other distance measures?
>
> 	- (Relevant content has been updated in the revised manuscript)
> 	- The Euclidean norm has proven to be a simple yet successful distance metric for measuring behavior differences in related literatures [1] [2] [3]. Intuitively, its quadratic form makes it sensitive for large deviations and can tolerate small displacements, which is a desired property in common cases. Another reason is that BEL uses TD3 as its base algorithm, which outputs deterministic actions rather than distributions. Due to these reasons, we assume it is a rational choose in our case. As a matter of fact, there are other interesting distance metrics used under different scenarios. For example, when the output actions are represented by distributions, the relative entropy can be used, when the action space is discrete, the Hamming distance can be used, and normalized compression distance is universal distance metric from an information theory perspective. You can refer to [4] and [5] for more in-depth discussions about this particular topic.
> - [1] Lehman, J., & Stanley, K. O. (2011). Abandoning Objectives: Evolution Through the Search for Novelty Alone. *Evolutionary Computation*, *19*(2), 189–223. [https://doi.org/10.1162/EVCO_a_00025](https://doi.org/10.1162/EVCO_a_00025)
> - [2] Mouret, J.-B., & Doncieux, S. (2009). Overcoming the bootstrap problem in evolutionary robotics using behavioral diversity. *Eleventh Conference on Congress on Evolutionary Computation (CEC’09)*, 1161–1168. [https://hal.archives-ouvertes.fr/hal-00473147](https://hal.archives-ouvertes.fr/hal-00473147)
> - [3] Plappert, M., Houthooft, R., Dhariwal, P., Sidor, S., Chen, R. Y., Chen, X., Asfour, T., Abbeel, P., & Andrychowicz, M. (2018). Parameter Space Noise for Exploration. *ArXiv:1706.01905 [Cs, Stat]*. [http://arxiv.org/abs/1706.01905](http://arxiv.org/abs/1706.01905)
> - [4] Doncieux, S., & Mouret, J.-B. (2013). Behavioral Diversity with Multiple Behavioral Distances. *IEEE Congress on Evolutionary Computation, 2013 (CEC 2013)*, 1–8. [https://hal.archives-ouvertes.fr/hal-01300703](https://hal.archives-ouvertes.fr/hal-01300703)
> - [5] Gomez, F. J. (2009). Sustaining diversity using behavioral information distance. *Proceedings of the 11th Annual Conference on Genetic and Evolutionary Computation - GECCO ’09*, 113. [https://doi.org/10.1145/1569901.1569918](https://doi.org/10.1145/1569901.1569918)

---

### Meta-Review · Area_Chair_4TyU · 2022-08-03

**Recommendation:** Accept (Poster)
**Confidence:** 4

**Metareview:**

Please check the comments of the reviewers in detail.

### Strengths
- the paper is well written and easy to follow
- experimental results (in simulation) are promising (better than SAC, TD3, and CEM-RL)

### Weaknesses
- motivation: the paper needs to explain better why diversity is likely to help
- definitions: the paper needs to explain better what the authors call a behavior and how they calculate a difference between behaviors
- weak statistics: only 10 replicates of the experiments, no statistical analysis (e.g. Wilcoxon test)
- weakly relevant to robotics: this is generic "RL for basic control" (Mujoco experiments) and does not address any specific challenge of current robotics
- the figures should be improved for readability
- several typos that should be easy to fix

Additional related litterature (not mentioned by the reviewers):
- behavioral sensitivity: Lehman, J., Chen, J., Clune, J., & Stanley, K. O. (2018, July). Safe mutations for deep and recurrent neural networks through output gradients. In Proceedings of the Genetic and Evolutionary Computation Conference (pp. 117-124).
- targeting behaviors: Baranes, A., & Oudeyer, P. Y. (2013). Active learning of inverse models with intrinsically motivated goal exploration in robots. Robotics and Autonomous Systems, 61(1), 49-73.

## Post-rebuttal update
I would like to thank the authors for their effort in improving the paper, which has been substantially improved.

---

> ### Author Response · Authors · 2022-08-22
> **Response to Area Chair 4TyU (Part1 / 2)**
>
> - We thank the reviewers and AC for their constructive and thoughtful feedbacks, and we appreciate their efforts in reading and understanding our work. We are excited to hear that our paper is easy to follow and the presented results promising. We'll address the issues brought up by reviewers and AC in the comments and try to further improve our manuscript as much as possible during the discussion period. The changes made to the manuscript will be colored blue.
> -
>   > motivation: the paper needs to explain better why diversity is likely to help
>
> 	- (Relevant content has been updated in the revised manuscript)
> 	- We apologize for not clearly stating our motivation, we think diversity is helpful in the following points:
> 		- Diversity is helpful for environments with sparse/deceptive reward signals where more diverse population have a higher chance of exploring potentially good states. [6]
> 		- Gathering a diverse set of experiences can help exploring more effectively and quickly, accelerating the learning procedure. [5]
> 		- High diversity is potentially helpful for multi-objective situations, e.g., training a cheetah robot to learn to both run forward and backward. [7]
> -
>   > definitions: the paper needs to explain better what the authors call a behavior and how they calculate a difference between behaviors
>
> 	- (Relevant content has been updated in the revised manuscript)
> 	- In BEL, the behavior is defined as the output actions evaluated on a batch of historical states sampled from the replay buffer. We assume this definition to be simple, with the ability to characterize how would an agent behave under different states.
> 	- The differences between agents' behaviors are calculated with the mean Euclidian distance between the action outputs of those agents evaluated on the same batch of historical states. The reason behind our choice of the Euclidean distance is discussed in the response to reviewer 1HqG.
> -
>   > weak statistics: only 10 replicates of the experiments, no statistical analysis (e.g. Wilcoxon test)
>
> 	- (Relevant content has been updated in the revised manuscript)
> 	- The decision of 10 repeated runs was done due to two reasons:
> 		- The setting of 10 repeated runs is common across related works involving off-policy algorithms, which are time-consuming: CEM-RL: 10 runs [4], PDERL: 5 runs [1], SAC: 5 runs [3], TD3: 10 runs [2]
> 		- Hardware limits. As we intended to replicate all comparing algorithms rather than copying results from other papers, a larger number of repetitions is hard to achieve in a reasonable time frame
> 	- We have updated our performance comparison table to demonstrate the advantage of our method with statistical significance by the Wilcoxon rank sum test. (Thank you for pointing out this!)
> -
>   > weakly relevant to robotics: this is generic "RL for basic control" (Mujoco experiments) and does not address any specific challenge of current robotics
>
> 	- We agree that our method does not tackle a concrete problem in robotics. We do have plans to further test and improve our method on real robot arms in the future.
> -
>   > the figures should be improved for readability
>
> 	- (Relevant content has been updated in the revised manuscript)
> 	- We apologize for the readability issues of the figures. In the updated manuscript, we have made improvements to them for a better reading experience.
> -
>   > several typos that should be easy to fix
>
> 	- (Relevant content has been updated in the revised manuscript)
> 	- Thank you for carefully reading our paper. We've further polished our manuscript to fix typos.

---

> > ### Author Response · Authors · 2022-08-22
> > **Response to Area Chair 4TyU (Part2 / 2)**
> >
> > - [1] Bodnar, C., Day, B., & Lió, P. (2020). Proximal Distilled Evolutionary Reinforcement Learning. *Proceedings of the AAAI Conference on Artificial Intelligence*, *34*(04), 3283–3290. [https://doi.org/10.1609/aaai.v34i04.5728](https://doi.org/10.1609/aaai.v34i04.5728)
> > - [2] Fujimoto, S., van Hoof, H., & Meger, D. (2018). Addressing Function Approximation Error in Actor-Critic Methods. *ArXiv:1802.09477 [Cs, Stat]*. [http://arxiv.org/abs/1802.09477](http://arxiv.org/abs/1802.09477)
> > - [3] Haarnoja, T., Zhou, A., Abbeel, P., & Levine, S. (2018). Soft Actor-Critic: Off-Policy Maximum Entropy Deep Reinforcement Learning with a Stochastic Actor. *ArXiv:1801.01290 [Cs, Stat]*. [http://arxiv.org/abs/1801.01290](http://arxiv.org/abs/1801.01290)
> > - [4] Pourchot, & Sigaud. (2018). *CEM-RL: Combining evolutionary and gradient-based methods for policy search*. [https://openreview.net/forum?id=BkeU5j0ctQ](https://openreview.net/forum?id=BkeU5j0ctQ)
> > - [5] Plappert, M., Houthooft, R., Dhariwal, P., Sidor, S., Chen, R. Y., Chen, X., Asfour, T., Abbeel, P., & Andrychowicz, M. (2018). Parameter Space Noise for Exploration. *ArXiv:1706.01905 [Cs, Stat]*. [http://arxiv.org/abs/1706.01905](http://arxiv.org/abs/1706.01905)
> > - [6] Hong, Z.-W., Shann, T.-Y., Su, S.-Y., Chang, Y.-H., & Lee, C.-Y. (2018). Diversity-Driven Exploration Strategy for Deep Reinforcement Learning. *ArXiv:1802.04564 [Cs, Stat]*. [http://arxiv.org/abs/1802.04564](http://arxiv.org/abs/1802.04564)
> > - [7] Parker-Holder, J., Pacchiano, A., Choromanski, K., & Roberts, S. (2020). Effective Diversity in Population Based Reinforcement Learning. *ArXiv:2002.00632 [Cs, Stat]*. [http://arxiv.org/abs/2002.00632](http://arxiv.org/abs/2002.00632)